# Pharmacological Prescription at the End of Life: Quality Assessment in the Transition of Care to a Community Palliative Care Support Team

**DOI:** 10.3390/pharmaceutics16091152

**Published:** 2024-08-30

**Authors:** Inês Rodrigues, Hugo Ribeiro, Carolina Costa, João Rocha-Neves, Marília Dourado

**Affiliations:** 1Community Palliative Care Support Team Gaia, R. Bartolomeu Dias 316, 4430-043 Vila Nova de Gaia, Portugal; 2Faculty of Medicine, University of Coimbra, 3000-548 Coimbra, Portugal; mdourado@fmed.uc.pt; 3Center for Innovative Biomedicine and Biotechnology—Group of Environment Genetics and Oncobiology (CIMAGO), FMUC, 3004-304 Coimbra, Portugal; 4Faculty of Medicine, University of Porto, 4200-219 Porto, Portugaljoaorochaneves@hotmail.com (J.R.-N.); 5Barão do Corvo Family Health Unit, 4400-035 Vila Nova de Gaia, Portugal; 6RISE@Health, 4200-219 Porto, Portugal

**Keywords:** pharmacological adequacy, palliative medicine, deprescribing, home care, polypharmacy

## Abstract

Appropriate pharmacological management is a cornerstone of quality in palliative care (PC), focusing on comfort and quality of life. Therapeutic review is crucial in PC, aiming to optimize symptom relief, reduce adverse effects, and manage drug interactions. This study aims to delve into the real-world pharmacological prescription practices within a Community Palliative Care Support Team (CPCST) in the northern region of Portugal, comparing practices at admission and at the last consultation before death. It is an observational, cross-sectional, retrospective study without intervention involving patients admitted to a CPCST in 2021. Data were obtained from clinical records, and the statistical analysis included descriptive and inferential measures. Sixty-four patients were included, with an average age of 77.34 years, referred mainly by a specialized Hospital Palliative Care Support Team (65.63%). Polypharmacy was present, with a significant increase in opioids, antipsychotics, prokinetics, antiemetics, antispasmodics, and local corticosteroids, and a reduction in drugs for peptic ulcer and gastroesophageal reflux treatment, antithrombotics, hypolipidemics, antihypertensives, and antidiabetics, among others. The oral route was preferred, decreasing between the two analyzed moments (85.5% versus 49.1%). Pro re nata (PRN) medications increased significantly (*p* ≤ 0.001). The prescription profile reflects a focus on symptom relief. The deprescription of drugs for chronic comorbidities suggests adaptation to care goals. At the end of life, PRN medications increase significantly (1.34 versus 3.26, *p* ≤ 0.001), administered as needed to soothe fluctuating symptoms. The pharmacological classes that have significantly increased are relevant in alleviating common symptoms in PC. The use of alternative routes for medication administration increases as instability of the oral route occurs, leading to a reduction in orally administered medications. Among these alternatives, the subcutaneous route shows the largest increase. The findings underscore the importance of flexible and responsive medication strategies in end-of-life care.

## 1. Introduction and Objectives

Community palliative care teams respond to patients with complex palliative needs through the direct provision of specific PC in the patient’s home, and support for their family members and caregivers [1,2].

The impact of symptoms on patients’ daily lives can be highly significant and harmful to quality of life. Patients with pain, dyspnea, and anxiety demonstrated an increasing worsening of their global functionality, reasoning ability, and difficulty carrying out simpler tasks than those they usually perform without these symptoms [3,4,5], so their control, through pharmacological and non-pharmacological measures, is a clear pillar of PC action [6,7].

Pharmacological prescription must be aligned with the defined care objectives [8]. It is expected that the prescription profile will reflect the importance of symptomatic control, especially in the most fragile and end-of-life patients, as the priority of treatment is to provide comfort and alleviate suffering [9,10].

In the last days or hours of life (LDHL), whose usual duration varies from hours to approximately 12–14 days, the most frequent symptoms and signs include delirium, rales, pain, dyspnea, nausea and vomiting [11,12]. Psychological symptoms such as anxiety, panic attacks, and episodes of intense fear are also common [11].

Polypharmacy is associated with an increased risk of falls, pharmacological interactions with unpredictable effects, worsening of patients’ general condition, and increased hospitalizations and mortality [13,14]. In Portugal, tools such as the BEERS Criteria and the STOPP/START criteria, adapted and translated into Portuguese, address this issue by helping clinicians identify potentially inappropriate medications and optimize treatment regimens [15,16]. It is also essential to avoid unnecessary polypharmacy, simplifying and suspending medications when the risks are more significant than the benefits, when there are more effective and safer alternatives or when the time needed for there to be a benefit is longer than the patient’s life expectancy [9,10,17]. Physiological changes, functionality and life expectancy and the patient’s preferences and beliefs must also be considered [18].

To mitigate symptoms of fluctuating intensity, pro re nata (PRN) medication is crucial in the treatment strategy [7,9]. PRN prescription was defined as a medication or treatment given only when it is necessary, rather than at regular intervals.

The primary objective of this study was to analyze the patterns of pharmacological prescriptions within a Community Palliative Care Support Team (CPCST), by comparing the prescription data at the time of patient admission with that of the final consultation prior to death.

The secondary objectives were to identify the main pharmacological classes and subgroups of drugs most used for regular and PRN administration, investigate the chosen drug administration routes and understand the changes in the prescription profile in the proximity of death.

## 2. Methods

The CPCST that collaborated in this study has the following cumulative admission criteria: residence in the team’s area of influence, diagnosis of chronic and progressive disease, symptomatic lack of control, need for collaboration in the organization of care, need for collaboration in the decision-making process, limitation of mobility and/or dependence (Palliative Performance Scale (PPSv2) ≤ 60%) [19] and existence of a capable caregiver [20].

Study design: Observational, cross-sectional and retrospective study without intervention. A classic literature review was also carried out using the terms mesh “Polypharmacy” and “palliative care” considering articles published from 2013 to 2023.

Selection of participants: We included patients admitted to the CPCST in 2021 and who were monitored for a minimum period of 48 h. The following patients were excluded: patients living in a residential structure for the elderly (RSE) or units of the National Integrated Continuous Care Network (NICCN), patients referred to other institutions/teams or clinically discharged from the CPCST, and patients with insufficient information on pharmacological prescription.

Data collection: Data were obtained through the electronic clinical process and the physical clinical file in the period of September–November/2023. The information extracted included sex, age, diagnosis, referring unit, main caregiver, level of functionality upon admission to the team (assessed using PPSv2), number of days of follow-up, drugs prescribed upon admission to the team (T0) and at the last consultation before death (T1)—pharmacological class, name of the medicine according to the international common name, route of administration and classification as regular medication or PRN medication. In the case of drug combinations of two or more drugs, each active substance was counted individually, except in combinations of laxatives and vitamin and mineral supplements.

The European General Data Protection act was respected and the identifiable data were anonymized and protected by a password.

Data analysis: Statistical analysis was carried out using SPSS (SPSS (IBM Corp., released 2021. IBM SPSS Statistics for Windows, version 28.0, Armonk, NY, USA).

The analysis involved descriptive statistics measures (absolute and relative frequencies, means and respective standard deviations, medians and interquartile range) and inferential statistics. The Spearman correlation coefficient, Student’s *t* test for paired samples, and the Mann–Whitney test were used. The McNemar test was used to study categorical variables. The significance level to reject the null hypothesis was set at α ≤ 0.05.

For analytical purposes and whenever considered appropriate, drugs were grouped into 28 pharmacological classes: opioid analgesics; non-opioid antipyretics and analgesics; nonsteroidal anti-inflammatory drugs (NSAIDs); antidepressants; anxiolytics, hypnotics or sedatives; antipsychotics; antiepileptics; anti-Parkinsonian; systemic corticosteroids; locally acting corticosteroids; drugs for the treatment of peptic ulcers and gastroesophageal reflux disease (GERD); prokinetics and antiemetics; antispasmodics; laxatives; antithrombotics; lipid-lowering drugs; beta blockers; antiarrhythmics; antihypertensives (excluding diuretics); diuretics; drugs used in benign prostatic hyperplasia; drugs used in obstructive respiratory disorders; systemic anti-infectives; antidiabetics (excluding insulins); insulins and analogs; vitamin and mineral supplements; oxygen; other drugs.

Ethical statements: This study was approved by the Local Ethics Committee (reference CE/2023/34) and the Helsinki Declaration was respected.

## 3. Results

In 2021, 166 patients were admitted and monitored by the CPCST for a minimum period of 48 h. Thirty-two patients were excluded because they lived in RSE or in an NICCN unit, sixty-three patients were excluded due to referral to other institutions/teams or due to clinical discharge from the CPCST, and seven patients were excluded due to insufficient clinical information on pharmacological prescription.

After applying the exclusion criteria, 64 patients were included in the analysis. The majority were male (54.69%). The average age was 77.34 years old, varying between a minimum of 38 and a maximum of 97 years old.

The length of follow-up by the CPCST had great variability, with a minimum of 2 and a maximum of 772 days. The median [IQR] was 27.50 days [7–49.5] and 26.56% of patients spent seven or fewer consecutive days with the team. 

Most patients (65.63%) were referred to the CPCST by a Hospital Palliative Care Support Team (HPCST). Regarding functionality upon admission to the team, 84.38% of patients had a PPSv2 score ≤ 40%, with more than half with scores ≤ 30%.

Table 1 shows the referring unit, main diagnosis of the evaluated patients, duration of follow-up, and functionality upon admission to the CPCST.

As for the main diagnosis, a high percentage (75%) referred to oncological diseases. The most common cancer was gastric cancer (18.75%), followed by colorectal cancer (16.67%). Dementia stood out clearly in diseases of the neurological system, representing 75% of cases. Of the organ failures, heart failure was the most common (62.5%). No secondary diagnoses or comorbidities were assessed.

In the therapeutic tables of the 64 patients under study, 136 different drugs were prescribed at regular times in T0 and 107 in T1 (for a total of 446 and 426 prescriptions, respectively). Regarding PRN medication, the count was 28 different drugs in T0 and 34 in T1 (in a total of 86 and 209 prescriptions).

Patients had, on average, 6.97 regular drugs prescribed at T0. This number decreased slightly at T1 (*p* = 0.602). On the contrary, the number of PRN drugs per patient increased significantly from T0 to T1 (1.34 versus 3.26) (*p* ≤ 0.001).

It was found that even considering only regular medication, 76.56% of patients met polypharmacy criteria (≥5 drugs) at T0, with this value reduced to 62.50% at T1.

There was statistically significant, positive, and weak correlation (*p* = 0.044) between PPSv2 upon admission to the team and the total number of regular drugs prescribed at that time. Thus, the higher the PPSv2 score (better functionality) at T0, the greater the number of regular drugs in the therapeutic table.

The correlation between PPSv2 at admission to the team and the total number of PRN drugs at that same time is not statistically significant (*p* = 0.307).

The most regularly used pharmacological classes in T0 were drugs for the treatment of peptic ulcers and GERD (8.74%), antidepressants (7.40%), and opioid analgesics (7.17%), while in T1 they were opioid analgesics (12.68%), prokinetics or antiemetics (8.45%), and anxiolytics, hypnotics, or sedatives (7.75%). The number of antispasmodics in therapeutic tables increased by 1800%. The number of lipid-lowering drugs reduced by 100%, antidiabetics (excluding insulins) by 89%, vitamin and mineral supplements by 80%, and antihypertensives (excluding diuretics) by 78%. Table 2 shows the drugs prescribed at T0 and T1.

From T0 to T1 there was a significant increase in opioid analgesics, antipsychotics, locally acting corticosteroids, prokinetics and antiemetics and antispasmodics and a significant decrease in drugs for the treatment of peptic ulcers and GERD, antithrombotics, lipid-lowering agents, antihypertensives, antidiabetics (excluding insulins) and vitamin and mineral supplements.

Furosemide topped the list of most regularly used medications, both in T0 (35.94% of patients) and in T1 (42.19%). Pantoprazole and lorazepam ranked second and third in T0, while fentanyl and ondansetron took second and third place, respectively, in T1. Other medications, such as butylscopolamine, morphine, budesonide, formoterol, and midazolam, gained prominence on the list of most used medications in T1.

Figure 1 shows the most prescribed medications at regular times in T0 and T1.

It was found that the oral route was the preferred route for administering regular medication at T0 and T1. However, while 85.5% of drugs were administered orally at T0, only 49.1% were administered via this route at T1. 

The transdermal (TD) route was the second most used in T0 (4.04%), followed by the inhaled route (3.81%). At T1, the subcutaneous (SC) route was the second most used, with 20.7% of regular drugs given via this route, and in third place was the inhalation route (11.7%). 

Only one patient had percutaneous gastrostomy (PEG) at T0, which was used to administer four drugs. At T1, two patients had PEG and 18 drugs were administered via this route. 

The intravenous (IV) route was not used for regular medication.

Table 3 shows the routes of administration of regular medication.

Upon admission to the team, at T0, no patient had a continuous subcutaneous infusion system (ISCC), while at the last consultation before death, at T1, this form of medication administration was used in 25 patients (39.06%). 

A total of 33 ISCC systems were used, as eight patients required two systems due to incompatibility of drugs administered through the same catheter. 

The five drugs most frequently administered for ISCC, in descending order, were: morphine (72% of patients with ISCC), butylscopolamine (60%), midazolam (56%), haloperidol (32%) and ondansetron (28%).

Opioid analgesics were the most used pharmacological class in PRN at T0 and T1, with a significant increase between the two moments. 

Between T0 and T1 there was also a significant increase in the PRN use of antipyretics and non-opioid analgesics, anxiolytics, hypnotics or sedatives, antipsychotics and prokinetics and antiemetics.

The prescription of NSAIDs in PRN decreased significantly, and these drugs were completely deprescribed.

Table 4 shows the drugs prescribed in PRN at T0 and T1.

Morphine and fentanyl were the opioid analgesics of choice in PRN. A total of 23.44% of patients used morphine at T0 versus 60.94% at T1, making it the most prevalent drug at both of the moments that were evaluated. Fentanyl was used by 15.63% of patients at T0 versus 28.13% at T1. 

Paracetamol was part of the PRN medications of 20.31% of patients in T0 and 53.13% in T1, being the second most prescribed PRN drug. 

Risperidone emerged significantly at T1, being prescribed to 46.88% of patients (compared to only 3.13% at T0). 

The most used routes of administration of PRN medications, both at T0 and T1, were the oral route, the rectal route, and the sublingual route, in descending order. 

It was found that 24.07% of PRN oral medication was administered in the form of drops at T0 and that this value rose to 69.67% at T1. 

The TD route was never used for PRN prescriptions and the IV route was used only once in T1.

## 4. Discussion

In our study, we included 64 patients with an average age of 77.34 years, most of whom were referred by an HPCST (65.63%). Polypharmacy was prevalent among the patients, with a significant increase in the use of opioids, antipsychotics, prokinetics, antiemetics, antispasmodics, and local corticosteroids, and a reduction in the use of drugs for peptic ulcer and GERD, antithrombotics, hypolipidemics, antihypertensives, and antidiabetics, among others, near the end of life. Although the oral route was preferred, its use decreased markedly. Additionally, the use of PRN medications increased significantly (*p* ≤ 0.001), with no differences observed related to referencing teams.

Oncological disease was the most common diagnostic category (75%). This finding is in line with data from other studies that show that professionals who specialize in PC still deal mainly with cancer patients [8,21,22,23,24,25,26]. However, it is known that the population suffering from non-oncological diseases requiring PC is increasing [2,24] and, therefore, the 25% of non-cancer patients cannot be devalued, as their needs are growing.

The fact that the average age of patients was high (77.34 years) was not surprising, as palliative needs increase in the elderly population, with a higher prevalence of cancer, cardiovascular diseases, dementia, and other chronic diseases [2,21]. However, the youngest patient was only 38 years old, emphasizing that PC is not exclusive to the geriatric population.

Patients were admitted to the CPCST with low functionality, given that 84.38% of patients had a PPSv2 score ≤ 40%. As one of the CPCST’s criteria is to monitor patients with a PPSv2 lower than 60%, these data are within expectations.

Patients had an average of 6.97 regular drugs prescribed upon admission to the team (T0) and 6.66 at the last consultation before death (T1). An older study involving cancer patients receiving outpatient PC found similar values [27].

The pharmacological classes that increased most significantly from T0 to T1 are relevant in the relief of frequent symptoms and, in addition, the drugs considered essential and appropriate by international experts [22,24,25,26,27,28] are among those most prescribed by the CPCST, which suggests a prescription aligned with the CP objectives.

It was found that 62.5% of patients had polypharmacy. Previous studies with patients at an advanced stage of disease who were receiving PC also reported high values [6,22,24]. In a study that included 100 patients with advanced cancer admitted to a PC or oncology service, 96% met polypharmacy criteria in the week before death and 61% were still taking more than five regular medications on the day of death [6]. 

In our study, the better the patient’s functionality upon admission to the team, the greater the number of regular medications in the therapeutic tables. Patients with better functionality present more favorable clinical conditions and are conducive to the active treatment of underlying chronic diseases, leading to a more intensive prescription of regular medication. On the other hand, new medications are introduced to alleviate uncomfortable symptoms [8,18].

The class of antispasmodics was the one in which the greatest increase was observed. Tatiana Peralta et al. also registered a significant increase in this class between admission to a Palliative Care Unit (PCU) and death [22]. Given the usefulness of antispasmodics in managing clinical manifestations of LDHL, a significant increase in their prescription was expected. Medications in this class, such as butylscopolamine, are used to alleviate sialorrhea and to reduce the volume of bronchial secretions and relieve rales [28]. 

Opioid analgesics were the most used pharmacological class in the last consultation before death, a result that is in line with data from other studies [22,25]. They can be essential in controlling more than one deleterious symptom, such as pain, dyspnea, and cough [11,28]. Of this group, the most prescribed medications were fentanyl and morphine, which was also similar to what was found by other authors [22,25]. 

Furosemide, which belongs to the class of diuretics, was the most regularly used medication at both moments that were evaluated. It can be used to control symptoms such as secretions in patients with evidence of pulmonary congestion, peripheral edema, and other symptoms resulting from exacerbation/decompensation of heart failure and cardio-renal syndrome [11]. It is voted among the top 10 drugs by a panel of experts questioned about essential medicines for patients in their last days of life, although no consensus at all has been achieved [10,29]. Considering the limitation of not having specifically assessed the need for the drug, the expectation regarding the use of furosemide in situations of pulmonary congestion, peripheral edema and other symptoms resulting from decompensation of heart failure and cardio-renal syndrome is congruent with the results obtained.

Medications such as fentanyl, ondansetron, butylscopolamine, morphine, budesonide, formoterol and midazolam stood out among the most used in T1. These are medications that are in line with the aim of relieving potential symptoms of LDHL, such as pain, dyspnea, nausea and vomiting, anxiety, terminal agitation, respiratory tract secretions, and rales [29]. Midazolam may also be useful in cases of palliative sedation [25,30]. 

A group of international PC experts reached agreement on four drugs relevant for symptomatic relief in LDHL: morphine (from the opioid analgesic class), midazolam (from the anxiolytics, hypnotics or sedatives), haloperidol (from the antipsychotics), and a drug with antimuscarinic properties [29]. All of them, except haloperidol, appear in this study on the list of most used medications in T1. Haloperidol, despite not being part of the ten most prescribed medications, was the most used antipsychotic in T1 and is among the five most prescribed drugs by ISCC. The results of an investigation conducted by Anniek D. Masman et al. corroborate these findings, highlighting haloperidol as the most prescribed antipsychotic in a PC center, as it is often administered SC in combination with other drugs [25]. This medication has been used to control agitation, confusion, delirium, and anxiety refractory to other pharmacological classes, also demonstrating efficacy as an anti-emetic [28,29], versatility that may explain its preference in symptom management. However, a randomized clinical trial with 247 patients did not demonstrate the superiority of haloperidol over placebo in controlling delirium [31].

All of the lipid-lowering drugs were deprescribed from T0 to T1. Antidiabetics (excluding insulins), antihypertensives (excluding diuretics), drugs for the treatment of peptic ulcers and GERD, antithrombotics and vitamin and mineral supplements also significantly decreased. This reduction in drugs intended to control chronic comorbidities reflects the change in care objectives, which are in line with the PPSv2 of the patients under study and their survival expectancy. In a study carried out in the context of a PCU in Portugal, most lipid-lowering, antihypertensive, and antithrombotic drugs were also discontinued [22]. In the face of advanced, progressive, and incurable disease, the risk–benefit ratio of pharmacological interventions changes. With increasing frailty and physiological changes due to the disease, drug interactions and adverse effects associated with medications may become more prominent and outweigh the potential benefits [24]. Near the end of life, the advantages of primary and secondary prevention, as well as active management of comorbidities, are less clear [24]. This is especially true for medicines which take a long time to exhibit beneficial effects, such as lipid-lowering drugs (namely statins), which are not appropriate when life expectancy is limited [9,17,22,24]. 

Current recommendations for the therapeutic approach to diabetes mellitus dyslipidemia, arterial hypertension, and heart failure, among other diseases, do not address the needs of patients with a life expectancy equal to or less than 12 months specifically or in sufficient depth. The lack of clear guidelines on therapeutic targets creates a gap in adapting treatments to this group of patients. Some patients receiving specialized PC maintain the use of medicines intended for primary and secondary prevention until the date of their death, with a likely increase in harmful effects and without additional benefit [24,32]. A 2021 literature review reports that, in studies with PC patients, the most frequently recognized potentially inappropriate medications (PIMs) are lipid-lowering, antihypertensive, antithrombotic, and drugs for the treatment of peptic ulcers and GERD [8]. According to the results of this same review, PIMs are a frequent cause of adverse drug effects, namely symptomatic hypotension, falls, hypoglycemia and blood disorders [8]. The results of the present study suggest that, despite not being noticeable due to the total number of drugs, the professionals involved were attentive and were active in deprescribing PIMs and simplifying therapy in situations of advanced disease and great fragility and complexity.

The number of PRN drugs per patient increased significantly between admission to the CPCST and the last visit before death. In patients at the end of life, it is good practice to anticipate the possibility of frequent symptoms, such as pain, nausea and vomiting, constipation, dyspnea, rales, anxiety, and delirium [23,24] and to ensure that there are PRN drugs for these relieve if necessary.

In our sample, there was a significant increase in PRN medication in the classes of opioid analgesics, antipyretics and non-opioid analgesics, anxiolytics, hypnotics or sedatives, antipsychotics and prokinetics and antiemetics, which supports the hypothesis that the intention is to safeguard patients from predictable suffering. Other authors, such as Russel et al., in a prospective study involving 203 patients, have reported increased PRN prescribing in the proximity of death [33].

Regarding regular drug therapy, despite the oral route being the preponderant route, there was a decrease in its use from T0 to T1. Swallowing difficulties and oral instability are common at the end of life, which may explain these results [24]. The second most used route of administration was TD at T0 and SC at T1. The results are in line with those of other authors who also identified the oral, SC, and TD routes as the most used in the PC environment and also recorded an increase in the SC route at the end of life [30,33].

In the last hours or days of life, drugs should not be started via TD due to the delay in onset and peak action, although those that are already in use can be maintained [24]. The SC route can be an excellent alternative to the oral route as it is comfortable, simple, and effective [24]. In this study, the non-use of the IV route for regular medication is probably related to the accessibility of the SC route, which has the advantage of being less iatrogenic [28,30].

This is particularly important as this CPCST provides care in patients’ homes, a less controlled environment than a hospital. Furthermore, this CPCST is not available 24 h a day, and since frail patients usually have poor venous access and hypovolemia, situations that lead to difficulty in placing and maintaining catheters, it would be expected that the IV route would have minimal use, as was demonstrated.

We cannot ignore that not all drugs are compatible with the SC route and many, despite being compatible, have been used off-label [30]. In any case, many relevant medications in PC are administered via this route, such as morphine, midazolam, haloperidol, butylscopolamine, furosemide, metoclopramide and ondansetron [28,30,31]. A total of 39.06% of patients included in this study used ISCC systems at the end of life, a figure very close to that found in a study carried out at a PCU in Portugal (40.9%).

The fact that the inhalation route is the third most used in T0 and T1 is influenced by the existence of medication that strictly requires this route of administration, introducing a bias into the statistical analysis.

Regarding PRN medication, in T1, more than half of the oral medication was in the form of an oral solution (69.67%), which was not the case in T0 (24.07%). This administration can be performed through drops. Drops, in addition to their convenience in allowing precise measurements, can be an affordable alternative for patients with difficulty swallowing.

## 5. Conclusions

Regular medication reviews are essential to ensure that each drug contributes meaningfully to symptom relief and aligns with the patient’s goals of care, focusing on improving quality of life. Far beyond deprescribing, it is important to adapt therapeutic strategies to the patient’s stage of life, taking into account the trajectory of the diseases that most affect them, as well as their organic and global functionality.

Although 62.5% of our patients have polypharmacy, the most important thing is to evaluate if they have adequate polypharmacy. This suitability and adaptation must not only take into account the objectives and place of care, but also the routes of administration and the doses and dosages used, with the main focus on obtaining the greatest therapeutic benefit and reducing the risk of adverse effects. 

In our study, we observed that patients had individualized care plans, with the use of the oral route preferably using oral solutions/drops at T1 (taking into account the loss of oral route that frequently occurs at this stage of life), the use of alternative routes (such as the transdermal route and subcutaneous route, with increases of almost 100% and 629%, respectively), and in regular medication, the focus is on controlling symptoms and quality of life (which is clear with the increase in prescriptions of 69% in opioids, 167% in antipsychotics, 320% in corticosteroids, and 1800% in antispasmodics, as the main examples). On the other hand, we observed the deprescription of medications related to the control of chronic diseases (for example, a reduction of 89% in antidiabetic drugs, 78% in antihypertensive drugs, 100% in lipid-lowering drugs, and 65% in antithrombotic drugs).

Regarding PRN medication, we observed a significant increase in prescription of opioids (about a 207% increase), antipyretics (about a 275% increase), anxiolytics (about a 417% increase), antipsychotics (about an 800% increase) and prokinetics and antiemetics (about a 317% increase). We concluded that this is a fundamental strategy to guarantee home care and home death for highly clinically complex patients.

## 6. Study Limitations

The analysis undertaken here did not verify the individual indications and benefits or harmful effects of each medication.

Comorbidities and physiological changes identified in patients were not evaluated. Indications for prescriptions, medication doses or medication adherence were not considered, and neither the symptoms experienced by patients nor their perception of quality of life were considered.

It is plausible to assume that the differences found between the two moments analyzed would be more pronounced if the majority of patients (65.63%) had not been referred by an HPCST, since the professionals who are part of HPCSTs, like CPCSTs, have specialized training in PC and tend to be more in tune with the importance of adapting medication prescriptions to meet the specific goals of PC patients.

Furthermore, we should consider the external validity of this study to other CPCSTs, as these are not homogeneous in terms of composition of the multidisciplinary team, training and specialization of professionals, availability of resources, operationalization of institutional policies, and characteristics of the population served, factors that directly influence clinical practice. The interpretation and application of the results of this study to other CPCSTs must consider the particularities of each team and the context in which they work.

## 7. Future Perspectives

Considering that the guidelines that guide pharmacological prescription in PC are mostly based on expert opinions [30], it is extremely important to continue studying this topic in a comprehensive way, not only to validate existing guidelines, but to identify areas that require further improvement. Knowing the dynamics of pharmacological prescription in PC, in the reality of clinical practice, can help to achieve an effective and compassionate response that serves patients and their families.

In the future, for a more representative view of prescription practices in home PC in Portugal, it is necessary to carry out multicenter registries involving CPCSTs across the country. Only by understanding the real-world practice can we effectively establish standardization of therapeutic attitudes, ensuring a consistent and high-quality approach.

## Figures and Tables

**Figure 1 pharmaceutics-16-01152-f001:**
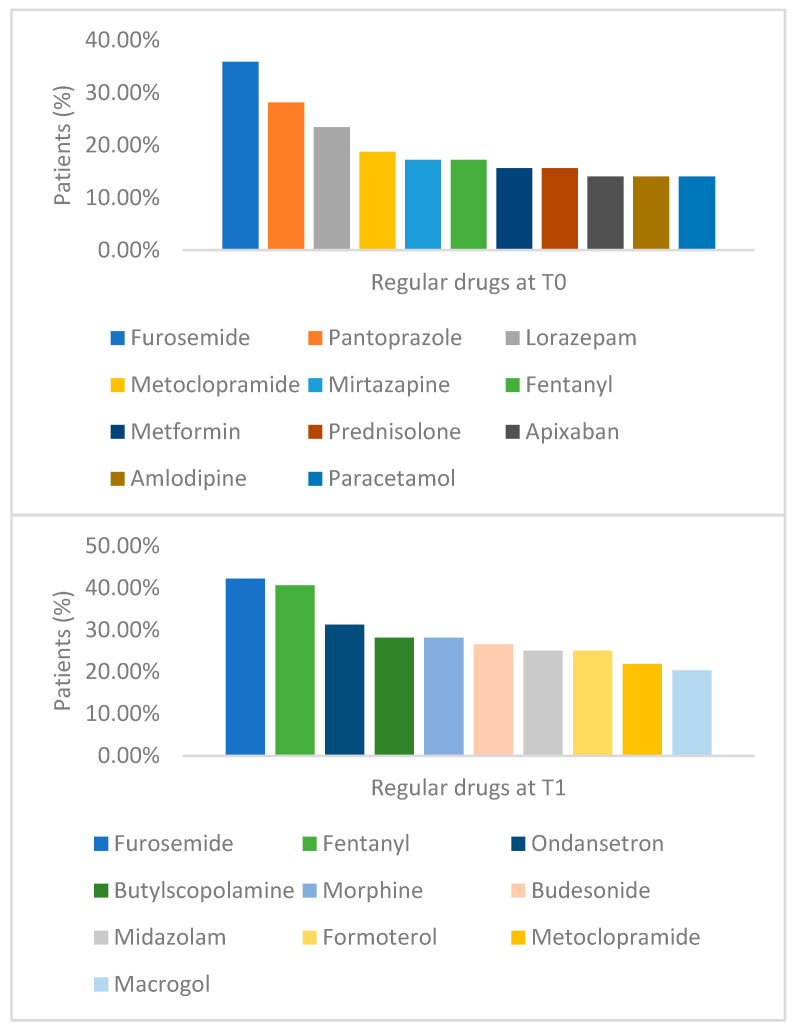
Medicines most used at regular times.

**Table 1 pharmaceutics-16-01152-t001:** Referring entity, diagnosis, functionality upon admission to the CPCST, and duration of follow-up.

	*N* = 64
Referring unit	*n* (%)
HPCST	42 (65.63%)
Family health team	21 (32.81%)
Other specialties	1 (1.56%)
**Diagnosis**	***n* (%)**
Neurological disease	8 (12.50%)
Dementia	6 (75.00%)
Parkinson	1 (12.50%)
Other	1 (12.50%)
Oncological disease	48 (75.00%)
Cancer of unknown primary	2 (4.17%)
Lung cancer	4 (8.33%)
Bladder cancer	1 (2.08%)
Breast cancer	3 (6.25%)
Prostate cancer	3 (6.25%)
Colorectal cancer	8 (16.67%)
Gastric cancer	9 (18.75%)
Pancreatic cancer	4 (8.33%)
Digestive tract cancer (other)	3 (6.25%)
Neurological system cancer	4 (8.33%)
Female genitalia cancer	1 (2.08%)
Head and neck cancer	1 (2.08%)
Synchronous cancers	5 (10.42%)
Organ failure	8 (12.50%)
Heart failure	5 (62.50%)
Hepatic failure	1 (12.50%)
Respiratory failure	2 (25.00%)
***PPSv2* at CPCST admission (%)**	***n* (%)**
10	4 (6.25%)
20	6 (9.38%)
30	23 (35.94%)
40	21 (32.81%)
50	8 (12.50%)
60	2 (3.13%)
**Follow-up duration (days)**	
***A* ± *SD***	57.30 ± 133.78
***Md* [IQR]**	27.50 [49.5–7.00]
**Min–Max**	2–772

*A*—Average; CPCST—Community Palliative Care Support Team; HPCST—Hospital Palliative Care Support Team; IQR—*Interquartile range*; Max—maximum; Md—median; Min—minimum; *n*—number; *PPSv2*—*Palliative Performance Scale*; *SD*—standard deviation.

**Table 2 pharmaceutics-16-01152-t002:** Differences in regular prescription between admission to the team (T0) and the last consultation before death (T1)–per dose.

	Number of Medicines		
Pharmacological Class	T0	T1	Percentage Δ
Opioid analgesics	32	54	+69%
Antipyretics and non-opioid analgesics	10	7	−30%
Nonsteroidal anti-inflammatory drugs	0	1	n/a
Antidepressants	33	19	−42%
Anxiolytics, hypnotics or sedatives	31	33	+6%
Antipsychotics	9	24	+167%
Antiepileptics	14	7	−50%
Anti-Parkinsonian	8	5	−38%
Systemic corticosteroids	17	13	−24%
Locally acting corticosteroids	5	21	+320%
Drugs for the treatment of peptic ulcers and GERD	39	21	−46%
Prokinetics or antiemetics	18	36	+100%
Antispasmodics	1	19	+1800%
Laxatives	20	25	+25%
Antithrombotics	23	8	−65%
Lipid-lowering drugs	7	0	−100%
Beta blockers	11	7	−36%
Antiarrhythmics	5	2	−60%
Antihypertensives (excluding diuretics)	27	6	−78%
Diuretics	31	29	−6%
Drugs used in BPH	9	8	−11%
Drugs used in obstructive respiratory diseases	14	32	+129%
Systemic anti-infectives	5	8	+60%
Antidiabetics (excluding insulins)	19	2	−89%
Insulins and analogs	6	3	−50%
Vitamin and mineral supplements	20	4	−80%
Oxygen	6	12	+100%
Other drugs	26	20	−23%

BPH—benign prostatic hyperplasia; GERD—gastroesophageal reflux disease.

**Table 3 pharmaceutics-16-01152-t003:** Routes of administration of regular medication–per dose.

	T0		T1	
	*n*	*%*	*n*	*%*
Oral	381	85.42	209	49.06
Transdermal	18	4.04	35	8.22
Subcutaneous	14	3.14	88	20.66
Inhalation	17	3.81	50	11.74
Nasal	6	1.35	15	3.52
Sublingual	0	0.00	4	0.94
Rectal	0	0.00	1	0.23
Ocular	5	1.12	3	0.70
Cutaneous	1	0.22	3	0.70
Intravenous	0	0.00	0	0.00
PEG	4	0.90	18	4.23

Legend: *n*—Number; PEG—percutaneous gastrostomy; %—Percentage.

**Table 4 pharmaceutics-16-01152-t004:** PRN prescription by pharmacological class, per patient, upon admission to the team (T0) and at the last consultation before death (T1).

	T0		T1		
Pharmacological Class	*A*	*SD*	*A*	*SD*	*p*
Opioid analgesics	0.44	0.50	0.91	0.39	**0.000**
Antipyretics and non-opioid analgesics	0.20	0.41	0.55	0.50	**0.000**
Non-steroidal anti-inflammatory drugs	0.06	0.24	0.00	0.00	**0.046**
Antidepressants	0.00	0.00	0.02	0.13	0.317
Anxiolytics, hypnotics, or sedatives	0.06	0.24	0.25	0.44	**0.003**
Antipsychotics	0.09	0.29	0.72	0.52	**0.000**
Antiepileptics	0.00	0.00	0.00	0.00	1.000
Anti-Parkinsonian drugs	0.00	0.00	0.00	0.00	1.000
Corticosteroids (oral and IV)	0.00	0.00	0.00	0.00	1.000
Locally acting corticosteroids	0.02	0.13	0.02	0.13	1.000
Drugs for the treatment of peptic ulcers and GERD	0.00	0.00	0.05	0.21	0.083
Prokinetics and antiemetics	0.06	0.24	0.19	0.39	**0.021**
Antispasmodics	0.00	0.00	0.02	0.13	0.317
Laxatives	0.30	0.71	0.30	0.46	0.766
Antithrombotics	0.00	0.00	0.00	0.00	1.000
Lipid-lowering drugs	0.00	0.00	0.00	0.00	1.000
Betablockers	0.00	0.00	0.00	0.00	1.000
Antiarrhythmics	0.00	0.00	0.02	0.13	0.317
Antihypertensives (excluding diuretics)	0.00	0.00	0.00	0.00	1.000
Diuretics	0.00	0.00	0.02	0.13	0.317
Drugs used in BPH	0.00	0.00	0.00	0.00	1.000
Drugs used in obstructive respiratory disorders	0.06	0.30	0.11	0.36	0.180
Systemic antibiotics	0.00	0.00	0.00	0.00	1.000
Antidiabetics (excluding insulins)	0.00	0.00	0.00	0.00	1.000
Insulins and analogs	0.02	0.13	0.05	0.21	0.157
Vitamin and mineral supplements	0.03	0.18	0.00	0.00	0.157
Oxygen	0.00	0.00	0.06	0.24	**0.046**
Other drugs	0.00	0.000	0.02	0.13	0.317

A—average; BPH—benign prostate hyperplasia; GERD—gastroesophageal reflux disease; IV—intravenous; *p*—significance; *SD*—standard deviation. Variations with statistical significance are highlighted in bold.

## Data Availability

Data available on considerable request.

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
