# Peer review of "Pharmacological Prescription at the End of Life: Quality Assessment in the Transition of Care to a Community Palliative Care Support Team"

_pharmaceutics, 2024, doi:10.3390/pharmaceutics16091152_

Round 1
Reviewer 1 Report
Comments and Suggestions for Authors
I read with interest the paper titled "Pharmacological prescription at the end of life: quality assessment in the transition of care to a community palliative care team".
This is an interesting paper in the quality of transition of care.
1. - line 60. Innapropriate prescription should be adressed in the background, with the explanation of the problem and the introduction of current tools used in Portugal to potential reduce the problem. As an example BEERS criteria or STOPP START translated to Portuguese tool could be referred as tools that could be used in this population to reduce the innapropriate prescription - This will improve your background.
2. Line 65 - Please describe the PRN concept. It's not cleary understood what it is, for the common reader of the manuscript.
3. Line 67 - what do you mean by "analyse the reality"? It means describe?
4. Methods - Its not cleary when the data was collected and what data refers to? It was collected in each year/month? Authors state that admission occured in 2021, but the data collection data was not referred.
5. Table 1 - Please refer which variables are exclusive or non-exclusive (eg: diagnosis)
6. Table 1 - Why some of the lines are highlighted in blue?
7. Table 2 - The data shared corresponds to the number of patients doing the medicine, or number of medicines used (for eg, if one patient takes two opioids, it counts as one or two?)
8. A suggestion is to present the figure 1 as side bars of moment T0 and T1 to understand the change.
9. Again, in table 3 is the count of medicines or patients? Please clarify within the table and discussion.
10. I would like to understand in the discussion, the thoughts of the authors about desprescribing. Polypharmacy in such settings are indeed a reality. Are all of them really needed?
Reviewer 2 Report
Comments and Suggestions for Authors
I agree with the authors that this study adds to the literature for transition to community palliative care (care at home), in particular as it relates to medication prescription patterns. Abstract, introduction, methods, analyses, results, discussion are all appropriately thorough. I agree with the limitations listed, in particular the applicability to other teams / other countries. The figures are well presented and clear.
Some changes that need to be made:
- all acronyms used need to be defined. Furthermore, there needs to be consistency in using the same acronyms for the same word. For example, the authors (who did not define either TD or DT) used TD/DT interchangeably and PCU/UCP interchangeably (also not defined) . They did not define EV or PIM (I still don't know what these mean) or ECSCP.
- some spellings need to be corrected: "hearth" (Table 1), "inalatory" (Table 3), "retal" (Table 3), "patintes" (line 360)
- line 106: the N needs to be capitalized in McNemar
- I'm not sure what "residual expression" means on line 378
- there should be consistency in terms used -- for example, is "bowel cancer" the same as "colorectal cancer"? In Table 1, "digestive tract cancer" is also used. In my mind, digestive tract includes the colon and rectum. How do you define it here? Evidently, given only 3 people are listed as having digestive tract cancer, your definition does not include the colon and rectum.
- line 54: "rales" are a sign (something physically observable), not a symptom (something the patient feels that is not physically observable, like dyspnea). Perhaps "...the most frequent symptoms and signs include..." would be a better way to phrase the sentence.
- unusual period placement (I wonder if these are typos or if it's a problem with how the journal's electronic system converted the original document to the PDF): line 181 (after "respectively"), line 309 (after "properties"), line 416 (after "improvement")
Comments on the Quality of English Language
see comments above
Round 2
Reviewer 1 Report
Comments and Suggestions for Authors
The overall quality of the paper was improved.
I just have two minor comments to add:
1. Reference 16 is citing a protocol for the translation - The correct reference is https://www.mdpi.com/1660-4601/19/11/6896
2. Instructions for authors should be followed in the abstract (no sections), Backmatter (as author contribution is missing, informed consent is missing, data availability statement is missing, institutional review board statement is missing). References are unformatted, and a conclusion section is also missing.
Author Response
Thank you for your attention and recommendations.
- Reference 16 is citing a protocol for the translation - The correct reference is https://www.mdpi.com/1660-4601/19/11/6896 - We corrected this reference.
2. Instructions for authors should be followed in the abstract (no sections), Backmatter (as author contribution is missing, informed consent is missing, data availability statement is missing, institutional review board statement is missing). References are unformatted, and a conclusion section is also missing. - we add author contribution, informed consent statement, data availability statement, institutional review board statement, and we included a conclusion section. We tried to correct the unformatted references, but we might need some technical support.
Reviewer 2 Report
Comments and Suggestions for Authors
I am satisfied with the revisions, which have addressed all of my previous questions and concerns.
Author Response
Thank you for your attention and suggestions.
Round 3
Reviewer 1 Report
Comments and Suggestions for Authors
Accept in the current form.
Author Response
Thank you for your time and attention.